# Underlying Motives for Selected Adventure Recreation Activities: The Case for Eudaimonics and Hedonics

**DOI:** 10.3390/bs10120185

**Published:** 2020-12-03

**Authors:** Alan Ewert, Ryan Zwart, Curt Davidson

**Affiliations:** 1Department of Recreation, Park, and Tourism Studies, Indiana University, Bloomington, IN 47401, USA; 2Outdoor Recreation Studies Program, Montreat College, Montreat, NC 28711, USA; ryan.zwart@montreat.edu; 3Recreation and Leisure Studies Department, California State University, Long Beach, CA 90032, USA; Curt.Davidson@csulb.edu

**Keywords:** adventure activities, eudaimonic and hedonic motivations, recreational risk-taking

## Abstract

One of the interesting behaviors practiced by citizens across the globe is the pursuit of outdoor recreational activities featuring elements of personal risk and danger. These types of activities are now becoming a global mainstay for many individuals, economies, and organizations. This study examined the underlying motivations and subsequent behaviors associated with risk-taking recreational activities and used the concepts of eudaimonics and hedonics to examine the motivations for participation from individuals engaging in three different adventure activities occurring in eight different locations. Recruitment took place in several forms, including in-person solicitation of participants at the activity areas, which consisted of mountain biking trailheads, rock climbing areas, and whitewater sites. Data were collected from three popular outdoor adventure activities (OAAs), including mountain biking, rock climbing, and whitewater boating. This study employed the use of multivariate analysis of variance (MANOVA) to investigate the relationship between two independent variable sets, including (1) the activity type, and (2) the level of experience, gender, type of activity, and the dependent variables of the Hedonic and Eudaimonic Motives for Activities (HEMA) scale (eudaimonic and hedonic). In addition, a cumulative odds ordinal logistic regression with proportional odds was utilized to determine the effects of expertise level and activity type on reported eudaimonic and hedonic motivations. A qualitative interview process was utilized to further investigate participant responses surrounding eudaimonic and hedonic motivational perspectives. The results indicated slight variations in experience level in the underlying motivations. Furthermore, qualitative inquiry revealed several motivation categories and diversity in the way those motivations were present throughout the recreation activity.

## 1. Introduction

The popularity of outdoor recreation activities is sustained, long-term, and critically important to economies and industry professionals. Defined as activities and experiences usually done in a natural or outdoor environment that involve elements of challenge and either real or perceived risk in which the outcome is uncertain, but influenced by the skill and ability of the participant [1], outdoor adventure activities (OAAs) often involve specific types of mental states, emotions, cognitions, perceptions, motivations, and associated behaviors that involve participation in these activities and experiences. This study examines the underlying motivations and subsequent behaviors associated with three specific OAA activities (mountain biking, rock climbing, and whitewater boating).

Motivation is defined as a process that initiates, guides, and helps maintain a goal-orientated behavior [2]. In addition, three components comprise the motivation complex: the activation of the specific motive, the persistence of that motive, and the intensity of that motive [2]. However, while motivations have been identified in the literature, researchers have found it challenging to structure these motives within high-order constructs [3]. This study sought to provide a somewhat new structure to those motives. Several studies have supported a connection between motivation and future behaviors. For example, McDavid, Cox, and Amorose [4] found motivation to be a strong determinant of self-reported leisure time physical activity behaviors. In addition, Kil, Holland, and Stein [5] reported a significant relationship between stated motivations and environment-based behaviors.

Of interest in this study was the casting of motives under the umbrella of hedonic and eudemonic perspectives [6,7]. Hedonic motives involve pleasure-seeking and comfort, while eudaimonic-derived motives are focused on factors such as self-expression or feelings of self-realization. This study used the concepts of eudaimonia and hedonics to examine the motivations for participation from individuals engaging in three different adventure activities in eight different locations. The hedonic and eudaimonic concepts of motivation were selected as the dependent variables in this study because, conceptually, they offer a different and more comprehensive picture of the complex phenomena of motivation.

Given the rapid growth of adventure-based recreational activities such as rafting, climbing, and adventure tourism, the results of this study will be useful in providing a deeper psychological understanding of the behaviors associated with the reasons why people engage in adventure-based behaviors. Accordingly, the following research questions guided this study:

RQ_1_: Do OAA participants take part in their activity due to eudaimonic or hedonic reasons?

RQ_2_: Are OAA activity types related to participant tendencies to report more eudaimonic or hedonic motivations?

RQ_3_: Does the level of experience of the participant influence eudaimonic versus hedonic motivations to participate in specific OAA activities?

RQ_4_: Does gender influence eudaimonic and hedonic motivations to participate in OAA activities?

## 2. Literature Review

### 2.1. Motivations

Understanding what motivates an individual to engage in a recreational activity that inherently involves elements of danger or potential risk has been the subject of a growing corpus of literature. Several theories have been developed that seek to explain the reason people engage in outdoor adventure activities. Early theories included arousal seeking [8], the peak experience [9], flow [10], and edgework [11]. Buckley [12] developed a classification system involving adventure recreation that uses the concepts of internal motivations, such as fear, control, skill development, and sense of achievement, and external motives such as social-based factors, defined as friendship development, image construction, escape, and competition with others or the environment. Expanding on this type of classification, this study specifically examined the relationship between hedonic and eudaimonic motivations within three different types of OAA activities, gender, and levels of experience.

Ewert and Hollenhorst [13] examined the relationship between the level of experience and the concept of internal and external motivations. Both Buckley [12] and Ewert and Hollenhorst [13] hypothesized that, as individuals become more engaged in a particular OAA activity, their motivations for participation will become more aligned with internal motivations, such as experiencing a personal challenge, achievement, control, and risk-taking, as opposed to external motivations, such as feeling pressured by friends or family to participate. Ewert, Gilbertson, Luo, and Voight [14] expanded on this by suggesting that as individuals develop skills and competencies concerning an activity, their intrinsic motivations for participation in this activity will continue to increase. Ewert et al. [14] also found that gender can exert an influential role in motivations for participation. In their study, females were more motivated by social considerations such as developing friendship, whereas males were more motivated by achievement and competition. Gilbertson and Ewert [15] also found support for the importance of gender and activity type, similar to the findings of Ewert, Gilbertson, Luo, and Voight [14], and that these reported motives stayed relatively constant over six years. Fisher [16] suggested that competence in an activity would not automatically increase intrinsic motivation in and of itself but needed to be coupled with a feeling of autonomy and positive impacts.

### 2.2. Eudaimonic and Hedonic Motives

Eudaimonic and hedonic concepts have a long history, originally being discussed as far back as ancient Greece [17]. More recently, the two concepts have often been associated with issues such as well-being and positive psychology. Within this context, hedonic motivations often consist of pleasure and positive emotions [18]. Several authors have posited that hedonic enjoyment refers to experiencing positive effects that accompany getting or having action opportunities that the individual wishes to experience [19,20]. Deci and Ryan [21] further elaborated that hedonic enjoyment can be felt whenever pleasant effects accompany the fulfillment of physical, intellectual, or socially based needs.

Moreover, eudaimonic aspects typically include facets of life meaning, personal growth and development, and fulfillment [22]. Huta and Ryan [23] suggest that while hedonically motivated activities are engaged in for pleasure and comfort, eudaimonic-motivated activities are engaged in by an individual to develop their best efforts. Huta and Ryan [23] further suggest that hedonic motives are related to subjective wellbeing (e.g., positive affect and carefreeness) while eudaimonic motives are more related to meaning. Waterman et al. [24] suggested that hedonic motivations are often short-lived and related to “getting important things that one wants” (p 235). Eudaimonic motives tend to be more complex and are often related to engaging in behaviors that are longer-lasting, deeply satisfying, and perceived to be worth doing [24]. In addition, Fletcher and Prince [25] posited that participating in an OAA for eudaimonic-based reasons may involve hedonic enjoyment, but not all hedonic-motivated activities involve eudaimonic motivations.

Eudaimonia is also thought to refer to feelings associated with developing an individual’s sense of fulling their potential and moving toward a sense of self-realization, life meaning, personal growth and development, and fulfillment [26]. Furthering this position, Ryan, Huta, and Deci [27] posit that eudaimonic benefits often stem from the satisfaction of the basic psychological needs of autonomy, competence, and relatedness, concepts strongly associated with self-determination theory.

It should be noted that there continues to be debate as to whether hedonic and eudaimonic motives are conceptually different. Kashdan, Biswas-Diener, and King [28] suggested that the two concepts are similar and often difficult to disentangle. Conversely, using a sample of outdoor education students, Vittersø and Søholt [29] and Della Fave et al. [22] found conceptual differences between the two types of motivations. Rounding out this disagreement, Henderson, and Knight [30] and Joshanoo [31] suggested that both hedonic and eudaimonic approaches denote important aspects of motivation. In this study, we take the position that hedonic and eudaimonic motivations are distinguishable enough to be measured, and this measurement forms the basis for this work and provides a more comprehensive understanding of motives for participation in OAAs beyond the more generalized internal and external motivations previously studied.

Within the OAA area, limited literature was identified that linked eudaimonic and hedonic motives with the specific independent variables of interest in this study, namely, gender, level of experience, and types of activities. For example, LeFebvre and Huta [32] assessed adults’ motivations in the pursuit of well-being across the adult lifespan. Their findings indicated that hedonic-based motives decreased for both males and females from their 30’s onward. For females, eudaimonic motivations increased until their 30’s and did not change significantly thereafter. For males, eudaimonic motivations decreased in their 30’s and 40’s, but then increased from their 40’s through 60’s. Houge, Mackenzie, and Brymer [18] suggested that in those activities using nature and the outdoors, eudaimonic motivations are emerging as more important than the previously assumed hedonic types of motivations. Related to this, Houge, Mackenzie, and Hodge [26] found that OAA activities foster eudaemonic aspects of subjective well-being by supporting the satisfaction of basic psychological needs for autonomy, competence, relatedness, and beneficence. Contrary to this finding, Zhou, Chleosz, Tower, and Morris [33] found more support for hedonic motives such as enjoyment, physical condition, and competition or ego in participants engaged in extreme sports and physical activity. This study examines the under-researched area of the relationship between eudaimonic and hedonic motivations for participation in OAAs concerning three specific OAA activities, gender, and level of experience. To date, little is known about the eudaimonic and hedonic perspective and the variables that influence these motivations within an OAA setting.

## 3. Methods

### 3.1. Participants

This study incorporated a quasi-experimental design. Recruitment occurred in several forms, including in-person solicitation of the potential participant at the activity area, such as mountain biking trailheads, rock climbing areas, or whitewater put-in sites. In-person participant solicitation occurred at areas proximal to the activity areas participants engaged in OAA would frequent, such as campgrounds, restaurants, and outdoor gear shops. Participants were approached at these locations and initially asked if they considered themselves to be a rock climber, whitewater paddler, or mountain biker. If potential participants agreed and were at least 18 years old, the researcher briefly explained the study to the potential participants. These data were gathered from sites in the southeastern and midwestern United States, including Kentucky, North Carolina, West Virginia, Michigan, and Indiana. All informed consent information was provided in the introductory pages of the web-based survey, and participants were able to discontinue participation at any time. During the survey, participants were asked to include their telephone number if interested in participating in a follow-up interview.

Participants agreeing to the study were randomly selected to participate in the semi-structured interview. Interviews ranged from 15–30 min long. The following are examples of the questions used in the interviews: In thinking about this activity, do you participate mostly for the excitement and challenge or for the meaning and personal value it provides you?When you think about this activity, do you mostly enjoy it when you are doing it or after it is over?When you think about this activity, do you engage for how you feel doing it or what it means to you afterward?When you think about this activity, which of these terms resonate with you: autonomy, mastery, personal growth, sense of achievement, or positive relationships with others?

### 3.2. Settings and Activities

Gilbertson and Ewert [15] reported that collecting data in an OAA setting can be particularly challenging for several reasons. First, OAA activities are often engaged in areas that are remote, subject to inclement weather and other conditions, are difficult for researchers to reach, and often include the wilderness and other backcountry locations. Second is the fluidity of the specific engagement. That is, participants are often focused on the upcoming activity, which can involve actual risk-taking. Moreover, if the participants are at the end of their engagement, they are often concerned with encroaching darkness or incoming weather and maybe physically tired. Thus, there is often a critical need for the researcher to be at a specific location to collect data. Depending on the actual situation, the quantitative data was collected in situ at the specific activity site prior, during, or directly after participating in the activity, while the qualitative data were collected via a phone interview at a later date.

In this study, three activities (rock climbing, mountain biking, and whitewater boating) were selected for two primary reasons. First, these activities represent examples of widely subscribed to OAA activities. For example, a 2019 report by the Outdoor Foundation [34] indicated that over 4.7 million people engaged in rock climbing, 5.9 million participants engaged in whitewater boating, and 8.7 million participated in mountain biking. Secondly, the researchers were all experienced in the three activities, and it was felt this experience might be useful in interpreting some of the responses within the qualitative portion of the analysis.

### 3.3. Procedures

After briefly explaining the study to potential participants, individuals willing to participate in the research were given a study information card that explained the study and contained both a uniform resource locator (URL) and a quick response (QR) code. Participants were asked to use the URL or QR code to access the study on their mobile device or personal computer. This gained the participant access to the online Qualtrics structure containing the survey. During the survey, participants were asked to include their telephone number if they were interested in participating in a follow-up interview at a later date.

### 3.4. Hedonic and Eudaimonic Motives for Activities Instrument

Participants were asked to complete the nine-item Hedonic and Eudaimonic Motives for Activities (HEMA) questionnaire [23,35]. The HEMA scale prompts participants by asking “To what degree do you typically approach your outdoor recreation activities with each of the following intentions, whether or not you actually achieve the aim?” Study participants then ranked the following motives on a 7-point Likert-type scale from 1 (not at all) to 7 (very much). Four motivations were eudemonic-based and included, “Seeking to pursue excellence or a personal ideal”, “Seeking to use the best in yourself”, “Seeking to develop a skill learn or gain insight into something”, and “Seeking to do what you believe in”. Five items with a more hedonic focus involved, “Seeking enjoyment”, “Seeking pleasure”, “Seeking fun”, “Seeking relaxation”, and “Seeking to take it easy”. Huta and Ryan [23] found the internal reliabilities of the HEMA scales by using Cronbach’s alpha coefficients of 0.82 for eudaimonic items and 0.85 for hedonic items. In addition, correlational analysis by Huta and Ryan [23] found that individuals who were more eudaimonic also tended to be more hedonic. This correlation will be discussed in a forthcoming section. A follow-up, semi-structured interview of a subset of the participant population provided a deeper understanding of participant perspectives, in addition to providing a more thorough understanding of these data.

### 3.5. Data Analysis

The following outlines the data analysis approaches for both the quantitative and qualitative phases of this study.

#### 3.5.1. Quantitative

Data were analyzed using SPSS version 27 (IBM, New York, NY, USA). For responses with missing data items, the researchers imputed a mean score for that item rather than removing the entire subject. Additionally, one respondent was deemed to be an outlier due to answering with the same response across all questions, and two other respondents did not provide any demographic information. All three respondents were removed from further analysis. The study first employed the use of multivariate analysis of variance (MANOVA) to investigate the relationship between the two independent variables (type of activity and level of expertise) and the dependent variable factors of the HEMA scale (eudaimonic and hedonic). A cumulative odds ordinal regression was then utilized to examine what effect expertise, activity type, and gender had on the reported eudaimonic and hedonic motivations.

#### 3.5.2. Qualitative

The purpose of the follow-up interviews was to further investigate participant responses surrounding eudaimonic and hedonic motivational perspectives. Interview subjects were selected randomly (with replacement) from a list of consenting respondents who had supplied their telephone numbers at the time of the initial survey. The telephone-based interviews lasted about 30 min, followed a semi-structured interview protocol, and incorporated in situ notetaking by recording the audio of the interviews for later review and transcription. The verbatim transcriptions were then used to analyze the data, using a constant comparison coding approach [36]. The time between meeting the respondent and interviewing the respondent was between one and two months.

In this analysis, the concept of saturation, or the point where the researcher begins to identify no new ideas or themes emerging within the data, was used to determine when it was acceptable to stop the analysis. The number of interviews conducted included 10 rock climbers, 10 whitewater paddlers, and 13 mountain bikers. Examination of the qualitative data for this project was intended to add to the understanding regarding the importance of hedonic and eudaimonic motivations for participation in OAAs.

## 4. Findings

Findings from these data were relevant for understanding what underlying motives contributed to individual participation in self-selected recreational activities having inherent elements of risk and danger. Results from these data suggested that the nuances in motivation for participation were consistent and subtle. A total sample size of *n* = 288 resulted in a breakdown of *n* = 92 participants who were rock climbers, *n* = 79 who were whitewater boaters, and *n* = 117 who were mountain bikers. In addition, the data resulted in 180 males and 103 females, with two identifying as other and three others not responding with a gender. Given the small number compared with the total sample, these three respondents were eliminated from the analysis. Women who completed the online survey comprised 31% of rock climbers, 27% of mountain bikers, and 34% of whitewater paddlers. Participants ranged in age from 18–69 years old. The mean age was 33 years old with a standard deviation of 11.9 years. A more in-depth description of participant sociodemographic information is available in Table 1.

### 4.1. Quantitative Results: MANOVA

Using MANOVA, there were no significant differences identified between the different types of OAAs (mountain biking, rock climbing, and whitewater boating) and the reported eudaimonic and hedonic motivational tendencies. In addition, there were no significant differences between participants’ levels of experience and reported eudaimonic and hedonic motivational tendencies. Finally, for the OAA participants in this study, no significant differences were using the variables of gender and reported eudaimonic and hedonic motivations.

While the quantitative analysis in this study overall rendered no statistical significance, analyses of the mean scores for the variable groups did provide some insight into the hedonic and eudaimonic tendencies of participants. Table 2 outlines the mean scores of each activity type and level of experience by the two dependent variables of the HEMA factors. Also in Table 2, participants exhibited slightly higher hedonic means than eudaimonic means for each independent variable group. Conversely, there was a slight change when considering the eudaimonic means. While not significant, males did provide slightly higher means (4.37 to 4.15) for hedonic motivations than eudaimonic. Similarly, females reported somewhat higher hedonic mean scores (4.31 to 4.19) than eudaimonic means when considering their OAA motivation. This finding appeared to diverge from many previous suggestions surrounding differences in gendered motivations in OAAs, where females generally reported higher levels of eudaimonic-oriented motivations when compared with males [14,15].

### 4.2. Quantitative Results: Regression

A cumulative odds ordinal logistic regression with proportional odds was utilized to determine the effect of eudaimonic and hedonic motivations among OAA participants. There were proportional odds, as assessed by a full likelihood ratio test comparing the fitted model to a model with varying location parameters (χ^2^ (180) = 74.933, *p* < 0.05). The deviance goodness-of-fit test indicated that the model was a good fit to the observed data (χ^2^ (3380) = 1052.415, *p* = 1), but most cells were sparse, with zero frequencies in 92.5% of cells. However, the final model significantly predicted one of the dependent variables over and above the intercept-only model: expertise (χ^2^ (12) = 21.4, *p* = 0.045). In this case, the data suggest that the level of experience, as a general construct, can affect different types of eudaimonic and hedonic types of motivation. This measure had a statistically significant effect on the prediction of whether someone reported hedonic or eudaimonic types of motivation (Wald χ^2^ (2) = 7.163, *p* = 0.028) and specifically that of being a novice. That is, a person who was a novice at the activity was more likely to report hedonic types of motivation than those with greater levels of experience. Additionally, when considering the participant’s level of experience, the quantitative data indicated a slight but consistent increase in eudaimonic motives from novice to intermediate and advanced individuals, suggesting that the greater the level of expertise, the greater the importance of eudaimonic-type motives.

### 4.3. Qualitative Results

The qualitative data collected for this study revealed some interesting trends and patterns that aided in the understanding of the quantitative results. First, these data confirmed that the difference between eudaimonic and hedonic motivations was subtle and, at times, difficult for the average adventure recreation user to parse out. During these interviews, subjects revealed that they participated in these OAA activities for a myriad of reasons, bouncing back and forth on the hedonic and eudaimonic spectrum. For example, users suggested that motivations differed at various time intervals throughout the activity. One mountain biker related that if the trail got harder, and the challenge for him increased, he was more likely to be experiencing hedonic types of motivations. However, after the experience, when he was recalling that experience to his social circle, he reported higher levels of eudaimonic types of motivations.

That led to perhaps the most interesting finding from the qualitative data, namely that hedonic types of motivations were usually expressed by users who were reporting their motivations during the activity. Conversely, eudaimonic responses were largely reported by users ascribing meaning from the experience after the activity, either recalling the experience on their own or in their social circles. Being able to reflect upon the experience and the presence of challenges seemed to be the two greatest predictors of how an individual spoke of their motivation for participation in an activity.

As stated, several sub-themes emerged from these data that could be categorized into eudaimonic or hedonic types of motivation (Table 3). These sub-themes helped describe the types of motivation experienced by the user, as well as illustrate what types of motivations they were experiencing throughout the recreation activity. For example, social connection was a common theme expressed by 65% of participants. One participant noted the following:

Climbing communities are great and it took me a while to find my community and once I did, it really changed my mind about climbing. Like at first, I was kind of putting myself down and then I met a really good community and I just wanted to climb every day.Participant RCK 2

Examples of meaning-making also permeated these data and included statements about timing, or when participants chose to reflect on the recreation experience. Participants identified the types of motivation they were experiencing both during and after the experience. These statements led the researchers to believe that timing of reflection was associated with what type of motivation an individual was likely to express. For example, more hedonic types of motivation were expressed during the activity, with statements like that from Participant MTB 3, who stated, “just thinking, like I guess when I’m doing it then like I can kind of just like be in the zone with like just pedaling and not like thinking about anything”.

These statements were juxtaposed with examples of meaning-making after the recreation activity, which tended to be more eudaimonic in nature. For example, Participant MTB 1 expressed, “And then the, you know, the stories and the experience you get to talk about afterward is enjoyable also. You know, did you see when I did that, did you see how this happened”.

Findings from the qualitative data also supported the results of the findings in the ordinal regression, in that users who were less experienced and had lower levels of experience tended toward more hedonic types of motivation. In the quantitative data, we observed a slight but consistent increase in reported eudaimonic motives from novice to intermediate and from intermediate to advanced levels of experience. This suggests that as individuals gain more skill and experience, there is a shift in their motivational patterns from hedonic to eudaimonic motivations.

## 5. Discussion

The findings of this study suggest that a user’s motivation may change with the level of experience and time (that is, whether they reflected on the activity during or after the activity itself). Moreover, these data suggest that both levels of experience and time for reflection result in a greater propensity to gravitate toward eudaimonic motives. We offer no definitive explanation, with the possibility that both greater experience and time for reflection allow participants to develop a better understanding of how the activity fits into their lives beyond a short-term, exciting experience.

Kahneman [37] offered a possible explanation, however, in the idea of two selves (the experiencing self and the remembering self). Kahneman and Riis [38] stated that the experiencing self is a part of the individual that is part of the moment cognitively, affectively, and behaviorally and responding to life as it exists. Zajchowski, Schwab, and Dustin [39] supported this contention and placed it into a leisure context. The remembering self is part of the individual that reflects on what they have experienced. Consequently, from a eudaimonic and hedonic perspective, when individuals consider the moment, they may have more of a hedonic perspective. When they think back on the OAA experience they tend to express a more eudaimonic motivation perspective. Thus, depending on whether an individual is thinking more immediately or more reflectively later, this may influence whether they will evaluate their motives as eudaimonic or hedonic.

The data from this study appeared to correspond with whether an individual was categorized as having higher values for hedonic or eudemonic motivations. Individuals who cited participation for the thrill of the activity, the risk, the danger of it, or the flow state they experienced were more likely to self-select into the hedonic category. Conversely, users that said they engaged in meaning-making after the experience were more likely to self-select into the eudaimonic category. The possible effect of timing has implications for researchers as they ascertain the participant’s self-reported meaning of the OAA experience, thus suggesting the need for reframing the way researchers examine motivation for future studies. More specifically, when data are collected, either before the engagement or after, should be considered an important factor in the design of future studies.

The quantitative results from the regression analysis revealed a change in motivation type depending on the level of experience. The data suggest that the more expertise in a specific outdoor adventure activity the user possesses, the more eudaimonic types of motivation they are likely to report. This finding is in line with several previous studies relative to the related terms of intrinsic motivation [11,18,20].

The qualitative data supported the findings from the quantitative analysis and additionally revealed several interesting themes that warrant further investigation. The variety of engagement, challenge, mastery, setting, social connections, connections to nature, and one’s ability to make meaning of the activity can all play a role in the complexity of the reasons for why people engage in a specific adventure activity and, once again, support the connection between motivation and direct behaviors. Future studies should consider the effects of a range of variables, such as those listed above, upon both the motives for participation and subsequent behaviors.

These findings can also factor into how OAA experiences are designed. For example, incorporating specific space and time for participants to make meaning from the OAA engagement can provide a higher quality experience. In addition, although gender did not have a significant influence, level of experience did. This suggests that organizations offering OAA experiences should consider the levels of experience among the program participants, again to offer a higher quality experience.

## 6. Limitations

Several limitations are associated with this study. First, as was described earlier, eudaimonic and hedonic motivations are often interlinked and present within the same time frame. Thus, participants may have had difficulties in their decision-making as to whether their motives for participation were of a eudaimonic or hedonic nature. Second, given the often physical and psychological intensity of the experience, accurately ascertaining the type of motive they were experiencing or did experience may have been cognitively challenging at the time for the participant. Third, the sample was restricted to three OAA activities, and it could be possible that other activities could be related to a different set of motives. To date, the literature provides little guidance for this possibility. Finally, a fourth limitation involved the disparity between male and female numbers, with the data in this study suggesting a gender breakdown of 66 percent male and 33 percent female. Although these data are in line with numerous other studies done in the OAA field [13,14,26], future research should involve a greater focus on a closer examination of female participants.

## 7. Conclusions

This study provided support for the interaction between eudaimonic and hedonic types of motivations during complex, high-adventure activities indeed being difficult for users to differentiate. The lack of significant mean differences suggests that users have a difficult time understanding their motivations, or have complex and often intertwined motivations, for engaging in the recreation activities they choose. Future studies should consider the complexity of the recreation activity experience and incorporate time and length of participation as additional variables for measuring motivations, as these seem to change throughout the activity for the individual.

## Figures and Tables

**Table 1 behavsci-10-00185-t001:** Participant sociodemographic data.

Variables	Activity Type		
	MTB	RC	WW
Age Range	18–69 Years	19–68 Years	18–63 Years
Gender	33 Females, 81 Males, 1 Other, 2 Preferred not to answer	42, Females, 48 Males, 1 Other, 1 Preferred not to answer	28 Females, 51 Males
Race	1 Other, 116 White	3 Asian, 2 Other, 1 Black or African American, 85 White	2 Asian, 3 Other, 74 White

MTB: Mountain Biking, RC: Rock Climbing, WW: Whitewater Boating.

**Table 2 behavsci-10-00185-t002:** Descriptive statistics mean comparison.

Variables	HEMA Factor	OAA Activity	Mean	Standard Deviation	*n*
Activity Type	Eudaimonia	Rock Climbing	4.16	0.64	92
		Whitewater Boating	4.27	0.63	79
		Mountain Biking	4.09	0.64	117
	Hedonia	Rock Climbing	4.32	0.46	92
		Whitewater Boating	4.38	0.46	79
		Mountain Biking	4.36	0.40	117
Level of Expertise	Eudaimonia	Novice	4.02	0.60	30
		Intermediate	4.15	0.60	120
		Advanced	4.20	0.68	137
	Hedonia	Novice	4.34	0.40	30
		Intermediate	4.35	0.46	120
		Advanced	4.36	0.43	137

**Table 3 behavsci-10-00185-t003:** Qualitative theme categories.

Eudaimonic	Hedonic
Social Connections	Flow State
Personal Growth	Sense of Mastery
Sense of Achievement	Challenge
Identity Work	Connection to Nature
Escape	Autonomy
Serious Leisure	Risk or Danger

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
