# Peer review of "Underlying Motives for Selected Adventure Recreation Activities: The Case for Eudaimonics and Hedonics"

_behavsci, 2020, doi:10.3390/bs10120185_

Round 1
Reviewer 1 Report
The article concerns difficult issues, touching the delicate field of human psyche. I appreciate the effort of the authors in understanding the motives of people undertaking various forms of recreational activity. My detailed comments relating to particular parts of the manuscript are as follows:
1. In the „Abstract", it is worth adding information about the results, about what was the most important finding
2. Keywords - I suggest not to duplicate the words used in the title of the work
3. Introduction and literature review
I have the impression that the introduction is too extensive, there are many references to literature, so I believe that the introduction can be combined with a literature review. This will make the text easier to communicate, highlighting research questions and perhaps raising hypotheses that are currently missing. I think that tables 1 and 2, which the authors refer to in the introduction, are unnecessary. You can include these data in the text. The literature review provides a good background for the formulated research questions. The first and third question is legible, clear, well put in the context of the literature review. The second question, however, makes me question. The authors explore only three types of outdoor activities. In my opinion, with such a small range of these activities, the search for differences due to the participants' tendencies to specific motivations may not be successful.
4. „Methods” chapter is divided into several sections. In my opinion, section one "Participants" - should contain the information presented in line 177-181. I consider the information about the age of the participants to be necessary, at least the average age. There were 288 people in the study, but gender data indicate 286, please verify this data. In studies in the field of psychology, equal numbers of gender groups are usually used. In these studies we have a visible male advantage. To what extent does this affect the results obtained? I think that the „Study design” part is redundant. Information from lines 181-183 should be included in the description of the „Procedure" (section 3.2). Table 3 is also superfluous. If you do not need to describe study sites more precisely, the information from Table 3 can be included in one sentence in the „Participants” section.
5. Results. I think that, in fact, qualitative analysis was very important to explain the research problem. It is only a pity that these results are based on a very small sample of participants. The data from Table 5 should be included in the text. This table contains not much information that will be easily understood if it is discussed in the text.
6. Discussion - this part should be completely rewritten. Lines 273-288 are again a literature review. Only the next paragraph is a reference to the results obtained. I miss the clearly defined limitation of this study. I appreciate the efforts of the authors in terms of practical implications of the results. But the statement in line 318 needs to be expanded. What do the authors specifically mean?
Thank you for allowing me to review this article. I hope that my suggestions and comments will be helpful for the authors.
Author Response
Responses to Reviewer 1
- In abstract most important findings have been added. We’ve also clarified the wording on the qualitative findings and fixed a few grammatical issues in the abstract. (Lines 11-29)
- Keywords – Our secondary literature reviewed the same key words as the original manuscript. We are comfortable with the keywords we have provided and are unaware of any other that should be included at this time. We are open to additional reviewer input on this issue should there be any as we may be using different spheres of literature as the reviewers. (Line 30)
- We did not combine the Introduction with the Literature Review but did shorten the Introduction. The data from Table 1 have been incorporated into the Introduction and Table 1 has been removed. The discussion of waves of research, including Table 2 have been removed from the manuscript. We felt this was unnecessary to the delivery of this project and think that it provides for a much clearer manuscript.
- Research Question 2 has been reworded to provide more clarity. (Line 66)
The second research question has been reworked to provide more clarity. (Lines 66-67)
A fourth research question has been added to account for the gender variable. (Lines 70-71)
In addition, a rationale for including only the three activities in the study has been provided. (Lines 191-197)
- The rationale for selecting the three OAA activities has been included. Our decision making on included these activities were access to sample and the idea that they provide some variety within the adventure recreation concept. We recognize this does not encapsulate the entirety of outdoor recreation. (Lines 191-197)
- The average age and age range has been included. These data are consistent with a wide range of studies and surveys done in the motivations area. (Lines 260-261)
- The difference in gender numbers has been clarified. (Lines 255-259)
- The visible male advantage has been addressed in the Limitations section of the paper. We think this was a good suggestion and provides for an important caveat when interpreting these data. (Lines 398-402)
- As per the suggestions by Reviewer 1, Table 3 has been removed and the information provided in the Participants section. (Lines 155-166)
- Information from Table 5 has been removed and the information contained in the text. (Lines 282-287)
- The Discussion has been rewritten to provide potential explanations as to the findings. We were careful not to extrapolate too much given the findings of the statistical analysis. However, we do think we added some strength to the findings with further interpretation of the research. (Lines 352-361)
- A Limitations section has been added to the text. Thanks for pointing out that we did not include this section. It’s important for us to provide this informative given the findings and considering the emphasis to future researchers that we make in the discussion section. (Lines 391-402)
- Material has been rewritten to provide more clarity relative to the practical implications of the findings. (Lines 382-389)
Reviewer 2 Report
Dear researcher,
Thank you for submitting your paper. The subject of your research is very interesting. So, I really encourage you to continue investigating it to further understand it.
Having said that, your research is promising but needs some work, specially the literature review and the results need to be exposed in greater depth. With the intention that you can make some improvements in your manuscript, I will comment on some of the most relevant weaknesses found in the paper. I hope you find those comments useful.
- The object of study. It would be interesting to know why did you choose those 3 activities and not others?
- Literature review. Authors need to present past literature on the study of these activities (how they have been study in the past) as well as the application of the hedonism and eudaimonia framework to hobbies / recreational activities. These aspects need further examination to understand the gap and the relevance of your effort and approach. Likewise, the importance of gender as a variable to analyze in the paper needs further explanation and support in past literature.
- Research questions. Following with my last comment, I am confused about the research questions. You say that “This study examined the motivations attendant to the fourth wave and specifically 84 investigated the relationship between hedonic and eudaimonic motivations within different types of OAA activities, gender, and levels of experience” but then you leave outside the gender issue.
Additionally, I wondered why different types of OAA activities and levels of experience can be relevant? You need to show that these variables are important or can affect the results, because it is not clear in the paper. By the way, if you examined other variables you should include in the manuscript as well. - Participants. I suggest including more information about the origin of the participants and where they were recruited. In this line, I would include more details about their sociodemographic profile and background (e.g. presented that information in a table). Why did you not analyzed/include other variables (e.g. social class, race/ethnicity, level of education...)?
Also, please explain with more details the information provided to the participants (card). You do not need to go in a lot of details, but you do need to give the basic structure of the information that you gave to the participants. After I wrote this comment I found out that you present this information under the section Study design (and a table)? Nevertheless, I would move most of this information to the participant section. By the way, how did you treat the respondent who chose the “other” option to the gender questions and the ones who did not answer the question at all (I see that you remove these)? - You need to explain the structure of the interview. Again, you do not need to go into a lot of details, but you do need to give the basic structure of the interview guide that you used in the process.
- I am very confusing about the structure you used in the Methodology section. I would suggest something like the following:
- Data collection and design:
- Participants. Survey (number of questions, structure... literature, etc.). Where did they take place? Any other details that can help the reader and future research?
- Participants. Guide for an interview. Time-length of the interviews. Where did they take place?
- Data analysis
- Quantitative
- Qualitative
- You need to prepare the reader about the methodology applied/used. It is confusing that in the abstract you only talk about the quantitative approach. Please revise and change this. This study employed the use of multivariate analysis of variance 20 (MANOVA) to investigate the relationship between two independent variable sets, including 1) 21 activity type, and 2) level of experience and the dependent variable factors of the HEMA scale 22 (Eudaimonic and Hedonic). Finally, a cumulative odds ordinal logistic regression with proportional 23 odds was utilized to determine the effects of expertise level, and activity type on reported 24 eudaimonic and hedonic motivations.
- Line 163: please correct the mistake “eudemonia”
- Line 214. Results from the data suggest that the nuances in motivation for participation are consistent 214 and subtle. This sounds more like a conclusion than a result.
- Quantitative results: Before jumping directly to the results of the MANOVA,you should present and comment on the results of the more descriptive analysis regarding eudaimonia and hedonic (table 4). This way we will know more about the motivations underneath the activities and not only about the differences between the activities. Furthermore, the descriptive analysis is especially recommendable in your case in which you did not find differences.
- Qualitative results. This section needs more work. What are the themes that appeared during the interviews? By the way, the text suggests that the subjects themselves defined whether their experiences were more hedonic/eudaimonic. If this is the case, in my opinion, it is a big mistake. First, not everyone knows how to differentiate the two concepts (even in literature there are problems/discussion); 2. the researcher does not control what the participant understands by such a concept. I would suggest go again to the data and analyze further under the lenses of positive psychology and other frameworks cited in your literature review. The results presented in the current paper are very superficial and insufficient, sorry. Please present the participants’ quotes to explain your finding. ;-)
- Regarding this finding “Conversely, eudaimonic responses were largely reported by users 261 ascribing meaning from the experience after the activity, either recalling the experience on their own 262 or in their social circles.” I would suggest that the literature on “experiential self” and the “remembering self” may help you to interpret your results.
- Table 2. Please revise it, it is difficult to understand. I would suggest you further develop its content and explain more in the tex.
- Tables: too much space for its content. I would suggest to move it to the text.
- You need to remove the text in lines 273-287 because it is a repetition of the introduction/literature review. For the data that is new, I would move it to the introduction/literature.
“Henderson and Knight [35] and Joshanoo [33] suggest that both hedonic and eudaimonic 273 approaches denote important aspects of motivation. Eudaimonia refers to feelings associated with 274 developing an individual’s sense of fulling their potential and moving toward a sense of self-275 realization [13]. In addition, eudaimonic aspects typically include aspects of life meaning, personal 276 growth and development, and fulfillment [27]. Huta and Ryan [28] suggest that while hedonically 277 motivated activities are engaged in for pleasure and comfort; eudaimonic-motivated activities are 278 engaged in by an individual to develop their best efforts. Huta and Ryan [28] further suggest that 279 hedonic motives are related to subjective wellbeing (e.g., positive affect and carefreeness) while 280 eudaimonic motives are more related to meaning.”
“Waterman et al., [13] suggests that hedonic motivations are often short-lived and related to 282 “getting important things that one wants” (p. 235). Eudaimonic motives tend to be more complex andare often related to engaging in behaviors that are longer-lasting, deeply satisfying, and perceived to 284 be worth doing [36]. In addition, Fletcher and Prince [37] posit that participating in OAA for 285 eudaimonic-based reasons may involve hedonic enjoyment but not all hedonic-motivated activities 286 involve eudaimonic motivations.
- Data collection and design:
- You say this in the discussion but it is not supported anywhere in your result section. The variety of engagement, challenge, mastery, setting, social connections, connections to 304 nature and one’s ability to make meaning of the activity all play a complex role in why an individual 305 engages in a specific adventure activity.
- This needs to go to the results section. For example, rock climbers who wanted a “casual day” were 306 likely to go out with a group of friends and climb easy routes. However, on a day they wanted more 307 “challenge” they were likely to go out with one individual of equal or similar abilities and climb at 308 the upper range of their abilities. Once again supporting the connection between motivation and 309 direct behaviors [38]. 310 The time at which a participant attempts to make meaning also seemed to correspond with 311 whether an individual was categorized as having higher values for hedonic or eudemonic 312 Individuals who cited participation for the thrill of the activity, the risk and danger of 313 it, or the flow state they experienced were more likely to self-select in to the hedonic category. 314 Conversely, users that said they engage in meaning making after the experience were more likely to 315 self-select into the eudaimonic category. The implications for these differences in timing are 316 interesting for adventure companies who might be inclined to facilitate a meaningful type of 317 experience for individuals. Further, this also factors into how recreation space is designed. One 318 implication would be to incorporate a meaning-making space where users can engage in these 319 sessions somewhere at the resource.
- Discussion You say that “This study confirmed that the interaction between eudaimonic and hedonic type of motivations 321 during complex, high adventure activities is indeed difficult for users to differentiate is indeed difficult for users to differentiate” but was “to know if users can differentiate them” the objective of the paper? Please revise this because it is very confusing.
Good luck with your research.
Author Response
Responses to Reviewer 2
Object of the study: The object of study. It would be interesting to know why did you choose those 3 activities and not others?
The rationale for selecting the three OAA activities used in this study are provided. (Lines 191-197)
Literature Review: Authors need to present past literature on the study of these activities (how they have been study in the past) as well as the application of the hedonism and eudaimonia framework to hobbies / recreational activities. These aspects need further examination to understand the gap and the relevance of your effort and approach. Likewise, the importance of gender as a variable to analyze in the paper needs further explanation and support in past literature.
The Literature Review has been strengthened to include specific information on motivations (eudaimonic/hedonic) and recreational activities. (lines 135-152)
In addition, we specify the gap in the research that this study attempts to address. (lines 149-152)
Material has been added regarding the relationship of level of experience and different types of motivations. (lines 84-99)
Extant literature describing the gender variable has also been added. The gap that this study attempts to fill is also described. (lines 92-97)
Research Questions: Following with my last comment, I am confused about the research questions. You say that “This study examined the motivations attendant to the fourth wave and specifically 84 investigated the relationship between hedonic and eudaimonic motivations within different types of OAA activities, gender, and levels of experience” but then you leave outside the gender issue.
Additionally, I wondered why different types of OAA activities and levels of experience can be relevant? You need to show that these variables are important or can affect the results, because it is not clear in the paper. By the way, if you examined other variables you should include in the manuscript as well.
The discussion regarding the different waves of research done in the area has been removed from the manuscript as being tangential to the intent of the manuscript.
In addition, more information has been added regarding the rationale regarding why those OAA activities were selected as well as attendant literature with the perspectives of gender and levels of experience. (lines 191-197, 92-97)
Participants: I suggest including more information about the origin of the participants and where they were recruited. In this line, I would include more details about their sociodemographic profile and background (e.g. presented that information in a table). Why did you not analyzed/include other variables (e.g. social class, race/ethnicity, level of education...)? Also, please explain with more details the information provided to the participants (card). You do not need to go in a lot of details, but you do need to give the basic structure of the information that you gave to the participants. After I wrote this comment I found out that you present this information under the section Study design (and a table)? Nevertheless, I would move most of this information to the participant section. By the way, how did you treat the respondent who chose the “other” option to the gender questions and the ones who did not answer the question at all (I see that you remove these)?
More information has been added regarding the demographics and origin of the participants. (lines 155-166)
A table (1) has been added that illustrates those demographic variables associated with the sample. (beginning with line 565)
A detailed account was added on how the respondents were approached, the information provided them, and specifics associated with the qualitative interview process. (lines 155-176)
Because of the low number of non-responses to the gender question, we chose to eliminate those responses from subsequent analyses. (lines 258-259)
You need to explain the structure of the interview. Again, you do not need to go into a lot of details, but you do need to give the basic structure of the interview guide that you used in the process.
Information on the interview process has been provided in lines 155-176, 199-205. We also provided a list of example questions here to further demonstrate the line of questioning utilized to initiate the conversations.
I am very confusing about the structure you used in the Methodology section. I would suggest something like the following:
ï‚· Data collection and design:
- Participants. Survey (number of questions, structure... literature, etc.). Where did they take place? Any other details that can help the reader and future research?
We have now provided much greater detail into the location and methodology used for the follow-up interviews. We’ve included time, place, location, and method for contacting the participants.
- Participants. Guide for an interview. Time-length of the interviews. Where did they take place?
- This information is all detailed now as a result of your comments.
ï‚· Data analysis
- Quantitative
- Qualitative
Data collection processes were expanded on and described in lines 199-221, 226-250. We provided details on the statistical analysis utilized as well as a number of additions to the ways in which the qualitative data were collected and analyzed.
ï‚· You need to prepare the reader about the methodology applied/used. It is confusing that in the abstract you only talk about the quantitative approach. Please revise and change this. This study employed the use of multivariate analysis of variance 20 (MANOVA) to investigate the relationship between two independent variable sets, including 1) 21 activity type, and 2) level of experience and the dependent variable factors of the HEMA scale 22 (Eudaimonic and Hedonic). Finally, a cumulative odds ordinal logistic regression with proportional 23 odds was utilized to determine the effects of expertise level, and activity type on reported 24 eudaimonic and hedonic motivations.
The abstract has been rewritten to provide more information regarding the findings of the study. (lines 25-29)
The “eudemonia” typo has been corrected.
Quantitative results: Before jumping directly to the results of the MANOVA,you should present and comment on the results of the more descriptive analysis regarding eudaimonia and hedonic (table 4). This way we will know more about the motivations underneath the activities and not only about the differences between the activities. Furthermore, the descriptive analysis is especially recommendable in your case in which you did not find differences.
Additional descriptive data has been provided in lines 252-262. In addition, a table (Table 2) of descriptive data has been added (beginning with line 578).
Qualitative results. This section needs more work. What are the themes that appeared during the interviews? By the way, the text suggests that the subjects themselves defined whether their experiences were more hedonic/eudaimonic. If this is the case, in my opinion, it is a big mistake. First, not everyone knows how to differentiate the two concepts (even in literature there are problems/discussion); 2. the researcher does not control what the participant understands by such a concept. I would suggest go again to the data and analyze further under the lenses of positive psychology and other frameworks cited in your literature review. The results presented in the current paper are very superficial and insufficient, sorry. Please present the participants’ quotes to explain your finding. ;-)
Themes garnered from the qualitative analyses have been added to the paper (lines 316-338). In addition, a table (Table 3) illustrating those themes has been added (beginning with line 585)
We have added material on the experiential/remembering self as an avenue of exploring why respondents reported the noted results. (lines 352-361). Thank you for the helpful insight!
Participant quotes have been provided when appropriate. We feel this was a good suggestion and will help provide rich context to the data. We utilized quotes to highlight our arguments and provide examples of the types of findings we’ve discussed.
Regarding this finding “Conversely, eudaimonic responses were largely reported by users 261 ascribing meaning from the experience after the activity, either recalling the experience on their own 262 or in their social circles.” I would suggest that the literature on “experiential self” and the “remembering self” may help you to interpret your results.
Literature on the experiential and remembering self has been added. Thank you, again for that suggestion, it was very helpful. (lines 352-371)
ï‚· Table 2. Please revise it, it is difficult to understand. I would suggest you further develop its content and explain more in the tex.
ï‚· Tables: too much space for its content. I would suggest to move it to the text.
ï‚· You need to remove the text in lines 273-287 because it is a repetition of the introduction/literature review.
Tables 1, 2, 3, and 4 have been removed and replaced with more descriptive tables. The introduction and literature review have been rewritten, in part, to avoid redundancy.
- For the data that is new, I would move it to the introduction/literature.
“Henderson and Knight [35] and Joshanoo [33] suggest that both hedonic and eudaimonic 273 approaches denote important aspects of motivation. Eudaimonia refers to feelings associated with 274 developing an individual’s sense of fulling their potential and moving toward a sense of self-275 realization [13]. In addition, eudaimonic aspects typically include aspects of life meaning, personal 276 growth and development, and fulfillment [27]. Huta and Ryan [28] suggest that while hedonically 277 motivated activities are engaged in for pleasure and comfort; eudaimonic-motivated activities are 278 engaged in by an individual to develop their best efforts. Huta and Ryan [28] further suggest that 279 hedonic motives are related to subjective wellbeing (e.g., positive affect and carefreeness) while 280 eudaimonic motives are more related to meaning.”
“Waterman et al., [13] suggests that hedonic motivations are often short-lived and related to 282 “getting important things that one wants” (p. 235). Eudaimonic motives tend to be more complex andare often related to engaging in behaviors that are longer-lasting, deeply satisfying, and perceived to 284 be worth doing [36]. In addition, Fletcher and Prince [37] posit that participating in OAA for 285 eudaimonic-based reasons may involve hedonic enjoyment but not all hedonic-motivated activities 286 involve eudaimonic motivations.
- This section has been rewritten, see lines 125-134.
- You say this in the discussion but it is not supported anywhere in your result section. The variety of engagement, challenge, mastery, setting, social connections, connections to 304 nature and one’s ability to make meaning of the activity all play a complex role in why an individual 305 engages in a specific adventure activity.
We reworded this and moved it into a section that talked about future research. (lines 378-383)
ï‚· This needs to go to the results section. For example, rock climbers who wanted a “casual day” were 306 likely to go out with a group of friends and climb easy routes. However, on a day they wanted more 307 “challenge” they were likely to go out with one individual of equal or similar abilities and climb at 308 the upper range of their abilities. Once again supporting the connection between motivation and 309 direct behaviors [38]. 310 The time at which a participant attempts to make meaning also seemed to correspond with 311 whether an individual was categorized as having higher values for hedonic or eudemonic 312 Individuals who cited participation for the thrill of the activity, the risk and danger of 313 it, or the flow state they experienced were more likely to self-select in to the hedonic category. 314 Conversely, users that said they engage in meaning making after the experience were more likely to 315 self-select into the eudaimonic category. The implications for these differences in timing are 316 interesting for adventure companies who might be inclined to facilitate a meaningful type of 317 experience for individuals. Further, this also factors into how recreation space is designed. One 318 implication would be to incorporate a meaning-making space where users can engage in these 319 sessions somewhere at the resource.
The above quote has been taken out and replaced with ones that was more productive (lines 322-325, 332-334, 337-338).
In addition the implications piece has been reworked to provide more clarity (lines 382-389)
Discussion You say that “This study confirmed that the interaction between eudaimonic and hedonic type of motivations 321 during complex, high adventure activities is indeed difficult for users to differentiate is indeed difficult for users to differentiate” but was “to know if users can differentiate them” the objective of the paper? Please revise this because it is very confusing.
We have reworded this in the Discussion section and included Kahneman’s work as a possible explanatory link (lines 352-361).
In addition, a Limitations section has been added. (lines 391-402)
Round 2
Reviewer 1 Report
Thank you for taking all my comments and suggestions into account. In my opinion, the text is now well developed and can be published. I believe it will be an interesting reading for the readers of the Behavioral Sciences.